# Skull Morphology, Bite Force, and Diet in Insectivorous Bats from Tropical Dry Forests in Colombia

**DOI:** 10.3390/biology10101012

**Published:** 2021-10-09

**Authors:** Leidy Azucena Ramírez-Fráncel, Leidy Viviana García-Herrera, Sergio Losada-Prado, Gladys Reinoso-Flórez, Burton K. Lim, Francisco Sánchez, Alfonso Sánchez-Hernández, Giovany Guevara

**Affiliations:** 1Programa de Doctorado en Ciencias Biológicas & Grupo de Investigación en Zoología (GIZ), Facultad de Ciencias, Universidad del Tolima, Altos de Santa Elena, Ibagué 730006, Colombia; lvgarcia@ut.edu.co; 2Programa para la Conservación de los Murciélagos de Colombia PCMCo, Bogotá 110911, Colombia; 3Departamento de Biología & Grupo de Investigación en Zoología (GIZ), Facultad de Ciencias, Universidad del Tolima, Altos de Santa Elena, Ibagué 730006, Colombia; slosada@ut.edu.co (S.L.-P.); greinoso@ut.edu.co (G.R.-F.); gguevara@ut.edu.co (G.G.); 4Department of Natural History, Royal Ontario Museum, 100 Queen’s Park, Toronto, ON M5S 2C6, Canada; burtonl@rom.on.ca; 5Grupo de Investigación ECOTONOS, Programa de Biología, Facultad de Ciencias Básicas e Ingeniería, Universidad de los Llanos, Villavicencio 500002, Colombia; fsanchezbarrera@unillanos.edu.co; 6Departamento de Matemáticas y Estadística, Facultad de Ciencias, Universidad del Tolima, Altos de Santa Elena, Ibagué 730006, Colombia; asanchez@ut.edu.co

**Keywords:** angle of opening, Chiroptera, length of skull, prey hardness, skull morphology

## Abstract

**Simple Summary:**

The cranial structure is highly variable among mammals and thought to reflect specializations for feeding and echolocation in bats. However, recent analyses of skull structure, feeding behavior and bite force across a wide range of bats suggest that correlations between morphology, performance, and ecology are not as delineated as previously thought. For example, most of the variations in bite force in insectivorous bats have been explained by differences in body size rather than specific cranial traits. We tested several relationships associated with the cranium to ascertain predictors of bite force in different bat species based on in vivo measurements from the Colombian tropical dry forests and museum specimens. Our data show that skull size had a significant contribution to bite force for beetle-eating bats, such as *Noctilio albiventris*, *Molossus molossus*, *M. coibensis*, and *Molossops temminckii*. Cranial traits and the combined action of the jaw morphology generate a biomechanical comparative advantage that allows these species to feed on “hard” prey, supporting the hypothesis that skull morphology, bite force, and diet are linked in insectivorous bats.

**Abstract:**

In Neotropical bats, studies on bite force have focused mainly on differences in trophic ecology, and little is known about whether factors other than body size generate interspecific differences in bite force amongst insectivorous bats and, consequently, in their diets. We tested if bite force is related to skull morphology and also to diet in an assemblage of Neotropical insectivorous bats from tropical dry forests in the inter-Andean central valley in Colombia. It is predicted that the preference of prey types among insectivorous species is based on bite force and cranial characteristics. We also evaluated whether skull morphology varies depending on the species and sex. Cranial measurements and correlations between morphological variation and bite force were examined for 10 insectivorous bat species. We calculated the size-independent mechanical advantage for the mandibular (jaw) lever system. In all species, bite force increased with length of the skull and the jaw more than other cranial measurements. Obligate insectivorous species were morphologically different from the omnivorous *Noctilio albiventris*, which feeds primarily on insects, but also consumes fish and fruits. Our results show that bite force and skull morphology are closely linked to diets in Neotropical insectivorous bats and, consequently, these traits are key to the interactions within the assemblage and with their prey.

## 1. Introduction

Many morphological characteristics of vertebrates are related to their diets, and this is associated with their skeletal structure and body size [1,2,3,4]. The cranium in particular responds to several selective pressures associated with food collection [5,6], respiration, and in some cases, structures to emit sounds related to feeding, reproduction or competition with congeners [7,8,9]. Differences in feeding habits correspond to cranio-morphological specializations [10,11], and these traits have been used to evaluate phylogenetic relationships [12], diet, and partition of food resources within vertebrates [13,14,15,16,17].

Bats are unique in that they are the only flying mammals and are the dominant aerial nocturnal vertebrates. In the Neotropics, there is a high diversity of species and trophic groups [18,19,20,21]. Their feeding strategies are associated with numerous physiological and morphological traits that facilitate the exploitation of food resources, and they have a diverse cranial morphology that appears to be related to changes in the bite force (BF) [22,23]. Studies on cranial morphology and BF have also shown evolutionary trends related to bat diets [6,24,25].

Differences in BF among taxa are related to the divergence of the skull shape and several ecological attributes, such as foraging, mating, and anti-predatory strategies [23,26]. BF is a performance trait and increased force can expand the range of prey available to a forager [27,28]. Bats with different feeding habits have distinctive skull morphologies and bite strengths that are linked to the ecological differentiation among species [25,28,29,30].

However, it has been argued that some species of insectivorous bats that can capture hard-bodied prey, such as beetles, prefer foraging on soft-bodied insects, such as moths [31]. Then, do bats select prey or simply eat whatever is available? This question is addressed by Kunz [32]; under the perspective of the optimal foraging theory, see also [33,34]. Food items eaten vary based upon availability of food and selectivity by bats. For most insectivorous bats, the “available” prey consists of all flying insects that bats are capable of capturing and eating during nocturnal activity. Insects consumed are also determined by several factors, including morphology of the predator, behavior of the prey item, ease of obtaining and using the item, size of the bat and of the prey, and nutritive value of the prey [32,33,35]. Therefore, it is not clear how skull morphology of insectivorous bats has developed to generate greater BF efficiency and greater success in capturing and consuming arthropods [10].

In light of all the above, we hypothesize that prey type among insectivorous bat species is based on BF and cranial characteristics [28,33]. In addition, we evaluate whether skull morphology varies depending on sex.

## 2. Materials and Methods

We used morphological data of bats from two sources: (i) a fieldwork in selected areas of the Colombian tropical dry forest, and (ii) vouchered specimens from biological collections; Museo de la Universidad Javeriana, Bogotá, Colombia (MUJ), Zoological Collection of the Universidad del Tolima, Ibagué, Colombia (CZUT-M), and Royal Ontario Museum, Toronto, ON, Canada (ROM).

### 2.1. Data Derived from Field Work

Fieldwork was done at 10 locations in areas of tropical dry forest (TDF) in Colombia (Figure 1). TDF is a globally vulnerable transitional biome between tropical humid forest and savannah ranging in elevation from 250 to 1000 m a.s.l., and is characterized by warm temperatures and marked seasonal precipitation [36,37,38]. The studied areas included heterogeneous landscapes of TDF, grasslands, vegetation in succession, and croplands [39], such as rice and other monocultures, in the department of Tolima (central Colombia; Figure 1, Appendix A and Appendix B). The sampling sites were separated by a minimum of 10 km to ensure the spatial independence of samples [40]. Captured bats were handled following standardized protocols for animal welfare [41], and approved by the Committee on the Use of Animals under the permit for collection of specimens endorsed by the Autoridad Nacional de Licencias Ambientales de Colombia-ANLA (resolution no. 02191 27 November 2018).

Between February 2019 and January 2020, bats were captured with mist-nets between 06:00 p.m. and 06:00 a.m. for six consecutive days per site. We used 3 mist nets, with a length and width of 12 × 2.6 m in the understory up to 3 m of height (sampling effort 216 h-net/day), 6 nets of 6 × 2.6 m in the sub-canopy between 6 and 9 m of height (sampling effort 432 h-net/day), and one “Triple High” net of 12 × 7 m [43] up to 7 m in height (sampling effort 72 h-net/day) in both rice fields and TDF remnants. After capture, age, sex, and reproductive status of each individual were registered, and only adult males and non-pregnant and non-lactating adult females were used in the study. The age of the bats were estimated based on the degree of ossification of the phalanges [44,45]. The reproductive status in the females was determined by examining the nipples and by palpating the abdomen [46,47].

We transported the captured bats in cloth bags, and BF of bats was measured using a portable digital fruit hardness tester Lutron FR 5120 (Taiwan) with a capacity of 196.10 Newtons and precision of ±0.05. In vivo measurements of the maximal bite force were recorded at the molars and repeated five times for each bat with at least 5 min between measurements, following the methods by Freeman and Lemen [48]. We considered the maximum value of the five measurements as the maximum BF produced by an individual. In addition, we also took 11 external measurements (see Table A1) using a Mitutoyo Absolute AOS digital caliper (accuracy 0.1 mm). All measurements were taken within 1 to 2 h after capture, and we subsequently released bats at the capture site. However, a minimum of seven individuals were collected for cranial measurements. All collected specimens were deposited in the Mammal Section of the Zoological Collection (CZUT-M) at the Universidad del Tolima (Ibagué, Colombia). We also measured prey hardness by using the Lutron portable tester (Model FR-5120).

During the fieldwork, we collected the feces of individual kept at least one hour in a cloth bag. The remains of insects found in the feces were compared with specimens from an entomological collection made during the fieldwork using light traps, Malaise traps, and yellow traps with pheromone lures to attract target moths known to act as agricultural pests: *Agrotis*, *Blissus*, *Diatraea*, *Mocis*, *Oebalus*, *Rupela albinella*, *Salbia*, *Spodoptera frugiperda* and *Tibraca*. We used specialized literature for the taxonomic determination of the insects to the most practical taxonomic level [49,50,51,52,53,54], usually morph species.

Within the prey items consumed by insectivorous bats, we also found seeds in the feces of *Noctilio*
*albiventris*, which were identified at the Dendrology Laboratory, Universidad del Tolima (Ibagué, Colombia). In addition, hardness measurements were made on 10 fruits taken in the field of the three identified plant species (Table A2) eaten by *N. albiventris* and we averaged the values for each plant species to estimate the hardness per fruit. This measurement was included in the global analysis of the food resources in the sampled areas. We measured the crude protein for two samples from each fruit and insect species and averaged the values. Each sample was 20 g and the crude protein content was determined by the centesimal composition method, following the recommendations by the Adolfo Lutz Institute [55] and analyses were performed at the Department of Chemistry, Universidad del Tolima (Ibagué, Colombia).

### 2.2. Skull Morphology

We examined skulls and mandibles of 528 adult bats (264 females and 264 males) belonging to 10 insectivorous species: *Peropteryx macrotis*, *Saccopteryx bilineata*, *S. leptura*, *Noctilio albiventris*, *Molossops temminckii*, *Molossus coibensis*, *M. molossus*, *Myotis nigricans*, *M. riparius* and *Rhogeessa io*. These specimens are deposited in the Royal Ontario Museum (ROM; Ontario, Canada), the Zoological Collection of the Universidad del Tolima (CZUT-M; Ibagué, Colombia), and the Museum of Natural History Lorenzo Uribe Uribe S. J.—Pontificia Universidad Javeriana (MUJ; Bogotá, Colombia). The localities of the specimens were obtained from the label of the voucher specimens and we georeferenced those that lacked coordinates.

For each specimen, we recorded 16 cranial measurements (Figure 2, Table 1) of the skull (11) and mandible (5). External measurements (Table A1) were obtained from the labels of the specimens recorded at the time of collection. Because there are differences in the size of the skull between species, the length of forearm (FA) was adjusted with a geometric mean [33,55,56,57]. All cranial measurements and BF were divided by FA [58].

### 2.3. Statistical Analyses

The 16 skull measurements used in the analyses were tested for non-collinearity by examination of bivariate scatterplots and correlation coefficients of r < 0.9. They were log_10_-transformed to meet the statistical assumptions of normality and homoscedasticity. A stepwise multiple linear regression model, discriminated by species and sex, was run with BF as the dependent variable and forearm (FA) and greatest skull length (GLS) as independent variables to explore which variable best explained the variation in BF. We evaluated whether the BF for each of the species was within the 95% confidence interval of the regression, and if it corresponded to the value predicted by the GLS. The tested model was Yijķ = µ + Ţi + δj + Ωķ + εijķ, where Yijķ represents the BF response at the j-th sex level and the i-th species; µ general average, Ţi effect produced by the i-th species, δj effect produced by the j-th sex, Ωķ effect due to the R-th trait of the 16 variables and εijķ the random error. We supplement our data by including information from other Neotropical bat species reported by Kalko et al. [59], and Marinello and Bernard [60]. We included FA and body mass to control for potential confounding effects related to body size. For this analysis, the logarithm was used to correct for violations in the assumption of normality.

#### 2.3.1. Variation in Skull Morphology

We analyzed the intra- and interspecific morphological variation of the skull and evaluated the possible effects associated with sex (Table 1) for each of the morphometric variables and the BF. The assumptions of normality and homoscedasticity were evaluated with the Shapiro–Wilk test and Levene’s test, respectively [61,62]. A multivariate analysis of variance (MANOVA) was done on these factor scores to test overall differences of cranial dimensions and BF between the selected bat species [61]. We then separated the data set according to sex, and used BF as the dependent variable, and use the species and sex as independent variables.

To explore the differences in skull morphology and body features between bat species and sexes, we performed a principal component analysis (PCA) based on a correlation matrix (Table A4). A threshold value of λ > 1 was used to determine relevant traits that explained the observed variance [62]. Based on the preliminary results of the PCA, we excluded the variable total length (TL) as it has a correlation coefficient near to 1. We also performed a canonical variate analysis (CVA) [62] on individuals with the 16 cranial measurements and the BF to establish the major axes of discrimination between the groups identified a priori, and to find the best linear combinations of variables with maximum discriminatory power between the resulting groups. Each discriminant group is represented by the vector of the means in all the variables to study the dimensionality of the data [62].

#### 2.3.2. Bite Force between Bat Species

The Kolmogorov–Smirnov test showed that our data did not fit a normal distribution, and thus we used nonparametric Kruskal–Wallis tests to compare the relative bite force (BF) after correcting for body size (BF/length of forearm, BF/cranial length, BF/cranial breadth, BF/cranial height) and external cranial dimensions (logarithmic transformation of the cranial length, width, and height).

#### 2.3.3. Bite Force, Skull Morphology, and Diet of Bats

We used a generalized linear model (GLM) with BF as a dependent variable, and cranial and external variables as independent ones. To complement the previous analyses, pairwise scatterplot diagrams based on a correlation matrix were constructed to describe the relationship among all cranial variables for identifying potential trade-offs or linear associations among variables. To verify if there was a statistically significant difference in the cranial variables and BF in comparison of males and females for each measurement, we used paired Student *t* tests (Table 2). All multivariate analyses and the graphical representations were performed in the program R version 3.5.3 [63], then post hoc tests were performed using Fisher’s least significant difference (LSD) pairwise comparison procedure. To evaluate the relationship between BF and species with different diets depending on the hardness of the prey or fruit resources in the case of *N. albiventris*, we used a basic data matrix (BDM) of 101 individuals belonging to 10 orders of insects found in the feces of bats (see Table A2). We performed a PCA with the BDM containing measurements of the insects (total length, wing length, hardness, crude protein, and ratio of crude protein: total length) and hardness of the fruits. All simple and multivariate analyses were performed in R version 3.5.3, using the packages “agricolae”; “multcomp”, “psych”, “FSA”, “ggplot2”, “car”, ”multcompView”, ”lsmeans”, “rcompanion”, “glm.predict”, “gamlss”, “glm.predict”, “gamlss”, “MASS”, “corrplot”, “ggpairs”, “FactoMineR”, “factoextra”, and “corrplot”) [62] with a significance level of α = 0.05.

## 3. Results

### 3.1. Morphological Variation of the Skull

The variation of cranial morphology was significantly affected by the species (F_5,216_ = 91.260, *p* < 0.01), the sex (F_1,216_ = 9.181, *p* < 0.01) and the sex by species interaction (F_1,37_ = 425.151, *p* < 0.01). The variables that significantly explained most of the variation among species (F_3,27_ = 4.79, *p* < 0.01) were: GLS, CIL, CCL, BB, ZB, DENL, and BF (Table 1 and Table A3). In general, cranial measurements were longer on average in females than in males (Table 1), and we found highly significant differences between all pairs of species (*p* < 0.001), except for the comparisons between *M. coibensis*—*M. molossus*, *P. macrotis—M. temminckii*, *P. macrotis—S. bilineata*, *M. nigricans*—*M. riparius* and *M. nigricans*—*S. leptura* (*p* > 0.05) (Table 3).

The PCA eigenvectors had the highest positive values for GLS, ZB, and M^3^–M^3^ on the first component, which indicates overall size (Figure 3, Table A5). Similarly, WMC and DD contribute the most to PC2, which indicates the shape of the jaw. The PCA indicated the presence of at least five groups: (1) *N. albiventris* is the most distinctive in size with the largest skull; (2) *M. molossus* and *M. coibensis* have moderate-sized skulls; (3) *M. temminckii* is similar in size to *S. bilineata* and *P. macrotis*, but has a higher bite force; (4) *M. nigricans* and *M. riparius* are similar in size to *S. leptura*, but have a higher bite force; and (5) *R. io* has the smallest skull. The three species in the sheath-tailed bat family Emballonuridae (*P. macrotis, S. bilineata*, and *S. leptura*) have a lower bite force than the others, which corresponds to smaller mandibular dimensions (Figure 3).

The analysis of variance of the morphometric variables indicated significant differences between the sexes (F_1,33_ = 9.19; *p* < 0.01) with females larger than males, including width across the third upper molars (M^3^–M^3^), greatest skull length (GLS) and zygomatic breadth (ZB). The results of the partial sum squares indicated that the cranial measurements and length of forearm (FA) differed between species and sexes.

The CVA using the twenty variables had 15 canonical vectors that were significant (*p* < 0.01) for discriminating only five species: *N. albiventris*, *Molossops temminckii*, *Molossus coibensis*, *M. molossus*, and *Myotis riparius* (Figure 4, Table A6). The first canonical function accounted for 79% of the variance (Wilks’ λ = 0.00013; χ^275^ = 1939.26, *p* < 0.01). The second canonical variable represented an additional 12% of the variance (Wilks’ λ = 0.0081; χ^2^_56_ = 1041.1, *p* < 0.01). The first two canonical axes did not recognize differences among *Saccopteryx*, *Myotis nigricans*, *Peropteryx*, and *Rhogeessa*. Molossidae bat species (*M. temminckii, M. molossus* and *M. coibensis*) together with *N. albiventris* appeared on the positive side of axis 1, whereas emballonurids and vespertilionids were on the negative side. The contribution of DD was highest on the first canonical axis, whereas WMC had the highest positive contribution and ZB the highest negative contribution on the second canonical axis.

### 3.2. Bite Force between Bat Species

All 19 cranial features were significantly associated with increased BF (*p* < 0.001), but length of forearm was not (Table 3). *N. albiventris* had the highest BF, followed by *M. coibensis*—*M. molossus, R. io, M. temminckii*, and *M. nigricans—M. riparius* (Table 1). The emballonurid bats (*S. leptura*, *S. bilineata*, and *P. macrotis*) had the lowest BF. Females had a stronger BF than males (*p* < 0.01) that was associated with greatest skull length (GLS).

Through the GLM model it was established that the bite force of insectivorous bats is influenced by all the cranial features studied here (*p* < 0.001, Table 3). It was identified by multiple comparison analysis that 16 traits are significantly associated with bite force, whereas the traits with the highest correlation were M^3^—M^3^, D—D, MTRL, PB, PL, and WMC regardless of sex (Figure 5).

Our analyses showed that *N. albiventris* was significantly different from the other species, as shown by post-hoc tests (*p* < 0.001). Additionally, we found that all congeneric species pairs were not significantly different from each other (Table 3). Moreover, there were significant differences in the BF of the bats depending on skull shape (Kruskal–Wallis, *p* = 0.012). The insectivorous bat species are different after taking into account sex (including GLS, rostrum length (PL), CCL, and M^3^-M^3^) and body size, but males and females of each species did not differ (W = 0.988, *p* = 0.004) (Figure 6). According to the pairwise comparisons of species (Table 3), BF behaved similarly in terms of variation in the shape of the skulls among paired taxa, except for *M. temminckii*—*P. macrotis*; *M. temminckii*—*S. bilineata*, *M. coibensis*—*P. macrotis*, *M. riparius*—*S. leptura,* and *P. macrotis*—*S. bilineata*, which were not significantly different.

We found a positive relationship between the BF of the different species and size of the skulls (GSL; T = 0.235, *p* < 0.01). Each value of BF and GLS (averaged from males/females) was significantly correlated by species based on linear regression (R^2^ = 0.7302, F_1,11_ = 34.0589, *p* < 0.001, Figure 7).

For the PCA (Figure 8) of insects and fruits consumed by bats, the total length and concentration of crude protein were the variables most correlated with PC1, whereas the width contributes the most to PC2. Insect taxa found toward the right of the plot are large, with high concentration of crude protein and low hardness (*Spodoptera frugiperda*, *Caulopsis microspora*, *Orphulella* sp., *Mocis* sp.), whereas taxa found on the left side are relatively small, with low concentration of crude protein, and high hardness values (insects: *Digitonthophagus* sp., *Oebalus* sp.2, *Onthophagus* sp., *Blissus leucopterus;* seeds: *Cecropia peltata, Cecropia obtusifolia* and *Piper cornifolium*). Insects in the upper right of the plot have relatively large wings (*Tagosodes* sp. and *Uvaroviella* sp.), and insects in the lower left of the plot have small wings (*Cerapachys* sp., *Aedes* sp. and *Thraulodes* sp.). All three species of seeds are hard and small.

## 4. Discussion

Differences in size between species, and in some cases between sexes, are dominant factors, but do not explain all the intra- and interspecific variations in bite force found among bats [22,64]. We show that an additional part of this variation can be attributed to differences in shape of the skull and jaw associated with large-scale changes in cranial length and small-scale changes to areas for muscle insertion. In particular, three measurements are associated with the development of the masticatory muscle [56,57]: width at mandibular condyles (WMC), dental depth under second lower molar protoconid (DD), and length of the masseter fossa length (MFL). By contrast, zygomatic breadth (ZB), greatest skull length (GLS), and width (M^3^—M^3^) are related to the size of the skull. This pattern is particularly evident in Emballonuridae, Vespertilionidae, and Molossidae because of extensive changes in shape and BF associated with specialization in insect consumption [28,33].

In a morphological context, the omnivorous *N. albiventris* does not completely escape the morphometric space of strict insectivorous bats. However, it did show a divergent anatomy in the PCA analysis associated with overall size (Figure 3). This is consistent with the consumption of a combination of hard insects [15], such as pieces of exoskeleton or hard shell (e.g., Coleoptera: *Digitonthophagus* sp., *Onthophagus* sp., Hemiptera: *Oebalus* sp.2, *Blissus leucopterus* [50]), and fruit with seeds (*Cecropia peltata*, *Cecropia obtusifolia*, *Piper cornifolium*). *Noctilio albiventris* also eats fish on occasion, but we did not find bones in the feces. Mounting evidence has suggested that a relationship exists between morphological variation and dietary diversity in bats [11,29]. The skull size of *N. albiventris* differed from aerial insectivorous bats in this study, which suggests that omnivorous insect and fruit eaters have broader mechanical ranges in their skulls, similar to those of insectivorous gleaners in cluttered spaces (*Gardnerycteris crenulatum*, Ospina-Garcés et al. [33]). Such convergence was the result of the adaptation of feeding apparatus to different diets, functional requirements, and morphological innovations that influence trophic performance [58,63,64,65,66,67].

The PCA separated bat species based on size, including the length of the skull (GLS), zygomatic breadth (ZB), and width across the third upper molars (M^3^—M^3^), which are traits associated with BF. However, the moderate-sized *M. molossus* could also be differentiated from the larger *N. albiventris* on PC2 (Figure 3) due to their wider mandibular condyles (WMC), masseter fossa length (MFL), and dental depth under the protoconid of m2 (DD), which are associated with the muscular attachments in the jaw and the BF. Although *N. albiventris* is about 25% larger than *M. molossus* based on greatest skull length (GLS), its bite force averages only 7% higher (Table 1). In addition, the emballonurid bats are separated from the other species on PC2, which corroborates their weaker bite force.

We also found that the shape of the skull, and not necessarily the size of the bat, was the factor responsible for the differences in BF in relation to sex. There was a trend for female insectivorous bats to develop a skull morphometry different from males [47]. This pattern has also been reported by McLellan [68], who found that males and females differ mainly in rostral width, depth of the braincase, mandibular width, palate length, and coronoid-angular distance in species of the fruit-eating bat genus *Carollia*. Furthermore, this pattern also is well documented among Vespertilionidae [69,70], and several species of Emballonuridae [71,72]. McLellan [68] addressed evolutionary explanations for sex differences in size, but our study suggests a different explanation might be necessary. It has recently been demonstrated that convergence in habitat specialists is not restricted to limb size and shape, but also occurs in other aspects of morphology, such as sexual size dimorphism and head shape [73]. Although functional demands imposed by prey traits, such as hardness and elusiveness may select for certain head shapes [28,33] in different sexes and age classes, it remains to be studied which aspects of the trophic niche may select for convergence in head shape in different ecomorphs.

The intermediate-sized insectivores in the sheath-tailed bat family Emballonuridae (*P. macrotis*, *S. bilineata*, *S. leptura*) have a lower bite force than the other species studied (Table 1). This result may have a trophic explanation because Herrel et al. [74] have shown that bite force is related to changes in the prey spectrum or prey types consumed. Our results also suggest that insectivorous bats with skulls specialized for a lower BF are those with smaller width across the mandibular condyles (WMC), masseter fossa length (MFL), and dentary depth at the second molar (DD), which are variables associated with the biomechanical action of the jaw.

The CVA identified six different groups with contrasting differences to the PCA because *Saccopteryx*, *Myotis nigricans*, *Peropteryx*, and *Rhogeessa* could not be discriminated. These results indicate that the primary difference in the depth of the dentary (DD) separate the other species from this group (Figure 4). This measure reflects differences in the size and development of temporal muscle fibers [75], and larger temporal fibers are related to increased body size and BF, due to a positive relationship with muscle mass [23,33,75,76,77]. This is consistent with studies such as Aranguren et al. [78], which proposed that *N. albiventris* can be considered a potential insect controller in disturbed areas close to dry forests in the Neotropics. The performance in the BF associated with the depth of the dentary and the biomechanical action of the jaw could be fundamental in the differences in foraging behavior, diet, and ecological niches within the sympatric species of *Noctilio*, *Molossus*, *Molossops* and *Myotis* [33,76,77,78]. Because limited habitat or food availability drives inter- and intraspecific competition [63], such morphological differences may optimize the use of resources and reduce diet overlap in various species, thus facilitating their coexistence [79]. For example, the morphological differences in two species pairs in *Artibeus* and *Carollia* affect their food choice and foraging behavior when feeding on fruits, resulting in ecological segregation in sympatric populations [80,81].

In the GLM, GLS and BF were significantly related in species with different diets as previously reported [22]. BF measured for *M. coibensis* fits the predictions for its size, as does that for *P. macrotis*, but the size-corrected BF in *M. coibensis* suggest differences in the performance of the chewing apparatus in this insectivorous bat. Moreover, size-corrected BF in *M. coibensis* (0.4, 0.32 N/GLS mm), was higher than in *Myotis riparius* (0.32, 0.31 N/GSL mm; [15]), but its value in *N. albiventris* (0.62 N/GSL mm), is similar to that of unrelated insectivorous bats of similar stronger bite force, such as *M. temminckii* (0.58, 0.58 N/GSL mm). Therefore, the differences in skull size and BF between insectivorous and omnivorous species are probably adaptations for trapping various insect species, such as beetles [15]. The temporalis muscle attachment and cross-sectional muscle area are related to body size with higher BF [82,83] in species with a larger cranium, a larger temporalis mass, and shorter temporalis fiber lengths [27], as found in our analysis.

The expected morphological similarities between phylogenetically related bat species are evident in *M. molossus* and *M. coibensis* [84,85]. A difference in the elevation angle of the condyle appears to have functional implications for insectivorous bats. A high opening angle is possible when the position of the temporalis muscle allows a higher moment around the temporomandibular joint (TMJ). In our study, this gape angle is related to muscle insertion on the masseteric fossa length (MFL), as has been demonstrated in mammals that feed on hard prey [10,86,87]. In our case, such hard prey/seed species were represented by *Digitonthophagus*, *Oebalus*, *Onthophagus*, *Blissus leucopterus*, *Cecropia peltata*, *Cecropia obtusifolia*, and *Piper cornifolium,* some of which are consumed by the omnivorous *N. albiventris*. The bats *N. leporinus* and *Myotis* spp. show a vertical temporal muscle and a high opening angle related to prey capture over water surfaces [88]. This angle indicates the orientation of the temporalis muscle [6,89], and determines the line of action of the BF [80]. A vertical orientation of the temporalis muscle of *N. albiventris* is associated with greater openness and less effort required to process food [90]. It has been argued that some species, such as *M. temminckii* and *P. macrotis* have low consumption of beetles due to the hardness of these prey and prefer soft-bodied insects, such as Lepidoptera, Orthoptera, and Diptera [31], with high concentrations of crude protein (e.g., *Spodoptera frugiperda*, *Caulopsis microspora*, *Orphulella* sp., *Mocis* sp. and *Aedes* sp.), contributing to the energy needed to fly long distances [90].

Our biometric analyses did not find cranial morphological similarities as expected among stronger bite force insectivores (*M. coibensis*, *M. molossus*, *M. temminckii*, *R. io*, *M. nigricans*, *M. riparius*, and *N. albiventris*) and weaker bite force insectivores in the family Emballonuridae (*P. macrotis*, *S. bilineata*, and *S. leptura*). The stronger bite force species differed in size and particularly in the angle of the masticatory apparatus and gape, measured in our study by the variable MFL. This suggests that there is not a clear association with the ingestion of insects between these species of bats.

## 5. Conclusions

Bite force and diet are linked in insectivorous bats with skull size playing a major role in determining a mechanism by which complex assemblages can partition food resources. Also, size is not the only factor involved and the mechanical advantage generated by the mandibular morphology gives insights into the trade-offs between bite force and biting speed that may influence prey selection, manipulation, and ingestion. A limitation of the present study was the relatively small number of species and narrow range of diets. Thus, more bat species and detailed food hardness data would be needed in a future study for supporting our results.

Additionally, the foraging strategy of some insectivorous species may also play an important role in diet. Further behavioral observations will also help in understanding the adaptations of bats to hunting insects in forested and agriculture-dominated areas in Colombia and other Neotropical countries.

## Figures and Tables

**Figure 1 biology-10-01012-f001:**
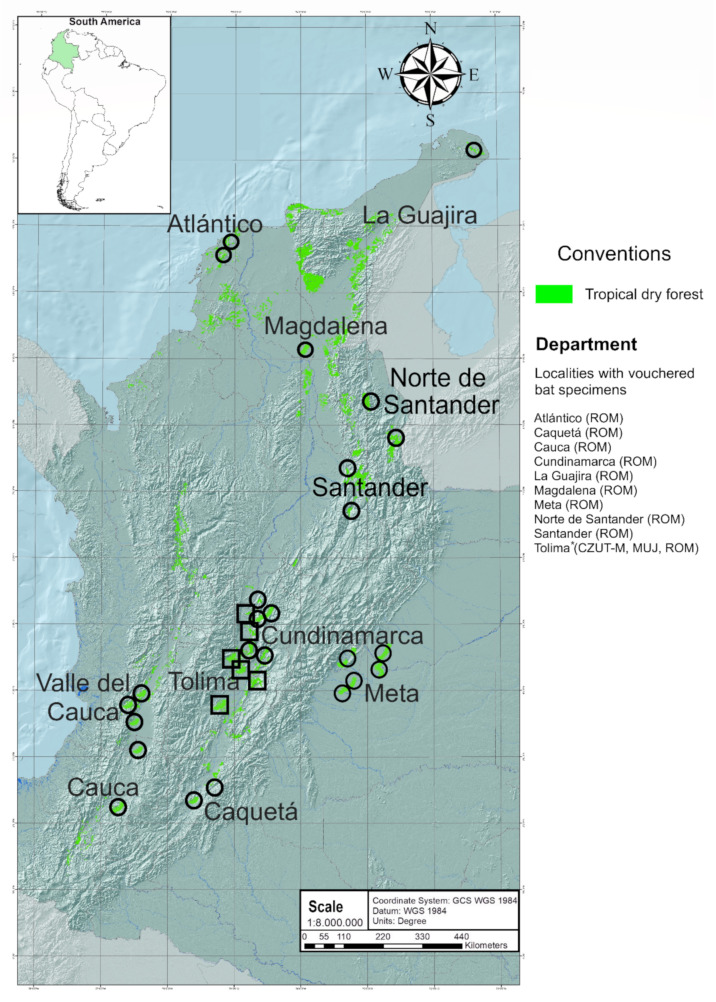
Sampling localities in 10 Colombian departments (regional division with fragments of tropical dry forest in light green following Ariza et al. [42] that had bat specimens used in our analyses. Circles indicate the sampling localities associated with vouchered specimens from museum collections, and squares indicate the sampling points in the Department of Tolima, which are specimens from fieldwork and deposited in the CZUT-M biological collection [(Ibagué, Colombia) (see Appendix B)].

**Figure 2 biology-10-01012-f002:**
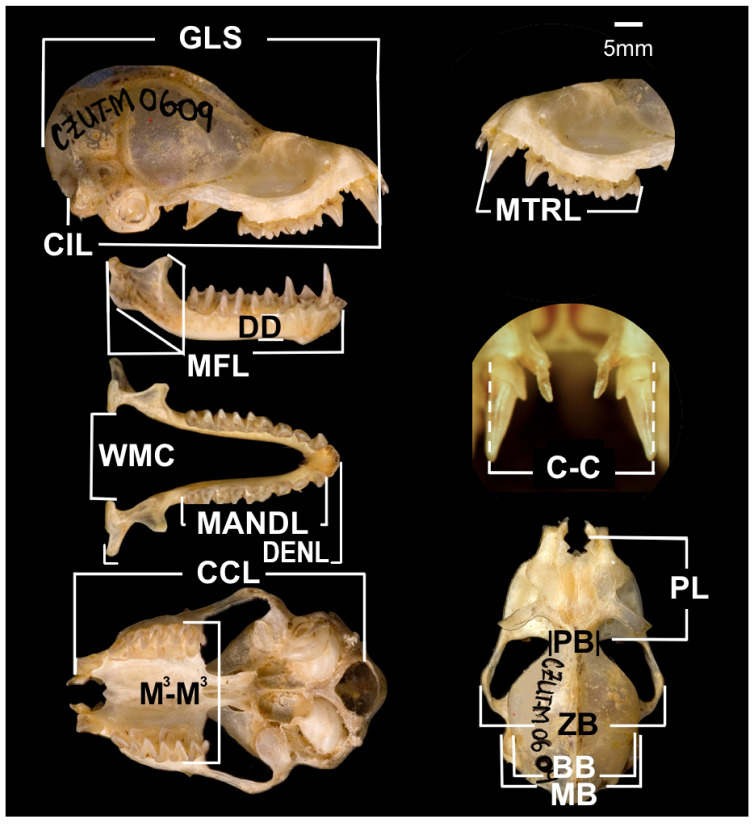
Skull and jaw measurements recorded in insectivorous bats from tropical dry forests in Colombia; *Saccopteryx bilineata* (adult male; scale: 5 mm). GLS, greatest skull length (excluding incisors); CIL, condylobasal length; CCL, condylocanine length; MTRL, maxillary tooth row length (C1-M^3^); PB, interorbital length; BB, braincase breadth; MB, mastoid breadth; ZB, zygomatic breadth; PL, rostrum length; M^3^–M^3^, width across third upper molars; C-C, palatal width at canines; DENL, dentary length; MANDL, mandibular tooth row length; MFL, masseteric fossa length; DD, dentary depth under the protoconid of the lower second molar; WMC, width at mandibular condyles.

**Figure 3 biology-10-01012-f003:**
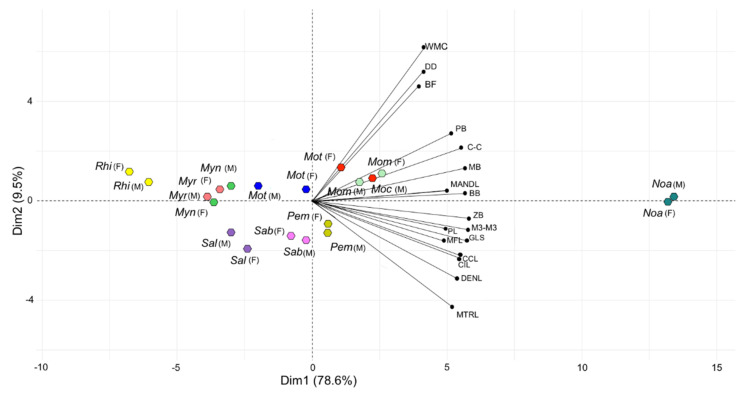
Principal component analysis with distribution averaged by species and sex of 16 craniodental morphological characters of 10 insectivorous bats recorded in tropical dry forests of Colombia. Abbreviations are showed in Figure 2 and explained in Table A1. Dimension 1 is mainly associated with the greatest skull length (GLS), width (M^3^–M^3^) and zygomatic breadth (ZB), whereas the second dimension is mainly associated with the width at mandibular condyles (WMC). *Peropteryx macrotis* (*Pem*), *Saccopteryx bilineata* (*Sab*), *Saccopteryx leptura* (*Sal*), *Noctilio albiventris* (*Noa*), *Molossops temminckii* (*Mot*), *Molossus coibensis* (*Moc*), *Molossus molossus* (*Mom*), *Myotis nigricans* (*Myn*), *Myotis riparius* (*Myr*), *Rhogeessa io* (*Rhi*).

**Figure 4 biology-10-01012-f004:**
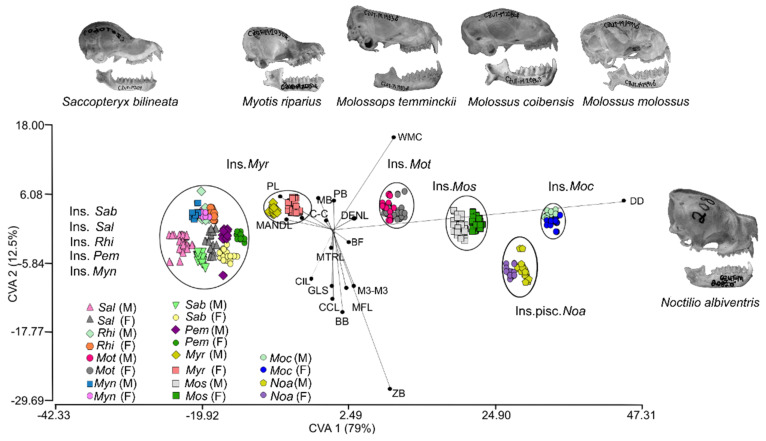
Canonical variate analysis of 16 craniodental morphological variables in 10 species of insectivorous bats (Ins.) separated by sex from Colombian tropical dry forests. The abbreviations of the variables correspond to those described in Table A1. Note that *Noctilio albiventris* is an insectivorous-piscivorous species (Ins. pisc.). *Peropteryx macrotis* (*Pem*), *Saccopteryx bilineata* (*Sab*), *Saccopteryx leptura* (*Sal*), *Noctilio albiventris* (*Noa*), *Molossops temminckii* (*Mot*), *Molossus coibensis* (*Moc*), *Molossus molossus* (*Mom*), *Myotis nigricans* (*Myn*), *Myotis riparius* (*Myr*), *Rhogeessa io* (*Rhi*).

**Figure 5 biology-10-01012-f005:**
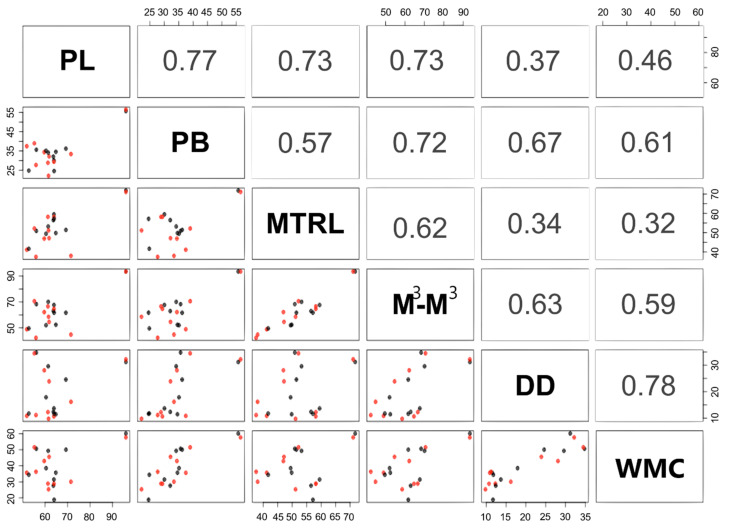
Scatterplot representing pairwise associations in the lower matrix and pairwise Pearson correlation coefficients in the upper matrix for 6 of 16 morphological variables from 10 insectivorous species from tropical dry forest areas in Colombia. Only the r’ values significantly different from zero (*p* < 0.05) are shown. PL = rostrum length; PB = interorbital length; MTRL = maxillary tooth row length; M^3^-M^3^ = Width across third upper molars; DD = dentary depth under the protoconid of the lower second molar; and WMC = width at mandibular condyles. Squares in the scatterplot are black for males and red for females.

**Figure 6 biology-10-01012-f006:**
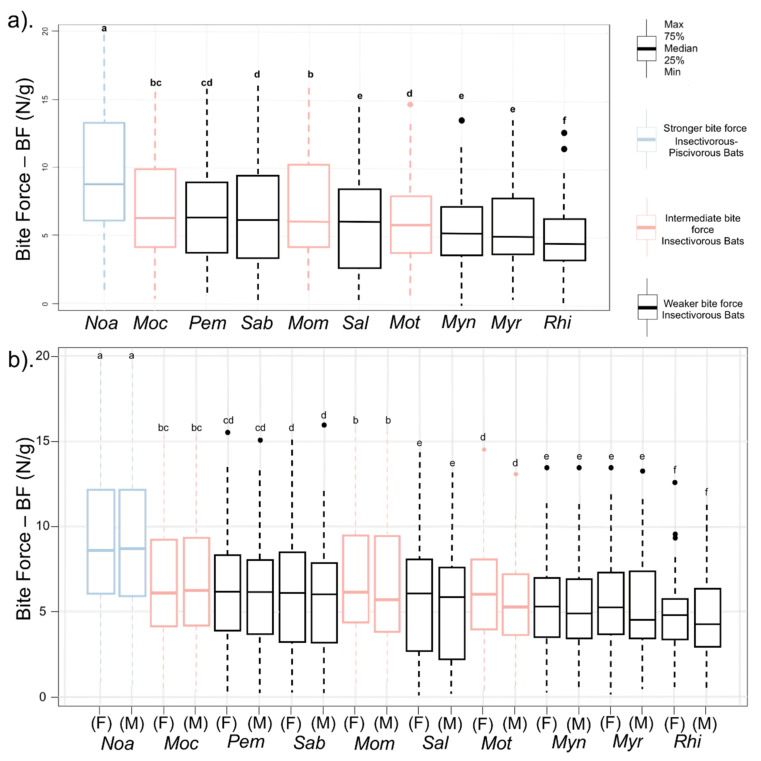
Differences in relative bite force of insectivorous bats from Colombian dry forest areas sorted among two diet categories corrected by their body sizes (BF in Newtons divided by the body mass (g) of the bats) with 95% confidence intervals resulting from the analysis of 16 craniodental variables: (**a**). Box plots with different letters indicating significant differences (LSD, *p* < 0.05), and (**b**). Separated by sex. *Peropteryx macrotis* (*Pem*), *Saccopteryx bilineata* (*Sab*), *Saccopteryx leptura* (*Sal*), *Noctilio albiventris* (*Noa*), *Molossops temminckii* (*Mot*), *Molossus coibensis* (*Moc*), *Molossus molossus* (*Mom*), *Myotis nigricans* (*Myn*), *Myotis riparius* (*Myr*), *Rhogeessa io* (*Rhi*).

**Figure 7 biology-10-01012-f007:**
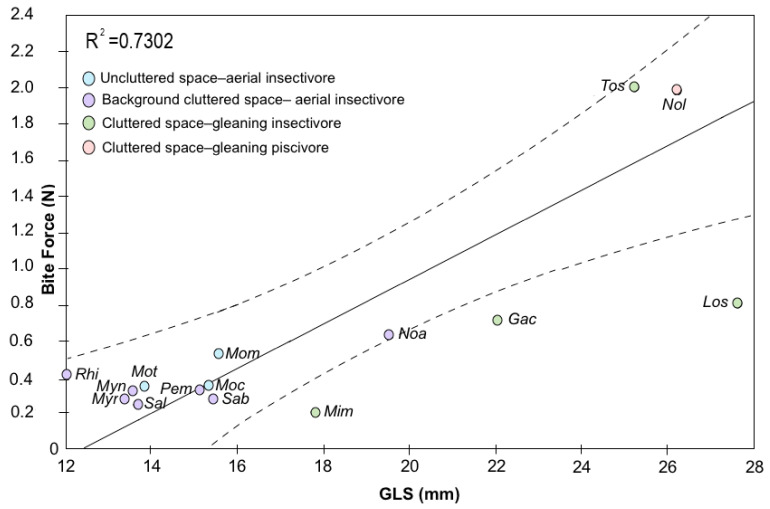
Linear regression showing that BF increases with GLS in insectivorous bats from tropical dry forests of Colombia (our data) and other Neotropical areas (data from Kalko et al. [59], and Marinello and Bernard [60]). Notice that most (3 of 4) phyllostomid bats fall below the predicted values of the regression. The dashed lines correspond to the 95% confidence intervals. Species were color-coded based on the 10 ecological guilds proposed by Kalko et al. [59], and Marinello and Bernard [60], which we sorted into four categories. *Peropteryx macrotis* (*Pem*), *Saccopteryx bilineata* (*Sab*), *Saccopteryx leptura* (*Sal*), *Gardnerycteris crenulatum* (*Gac*), *Lophostoma silvicolum* (*Los*), *Micronycteris microtis* (*Mim*), *Tonatia saurophila* (*Tos*), *Noctilio albiventris* (*Noa*), *Noctilio leporinus* (*Nol*), *Molossops temminckii* (*Mot*), *Molossus coibensis* (*Moc*), *Molossus molossus* (*Mom*), *Myotis nigricans* (*Myn*), *Myotis riparius* (*Myr*), *Rhogeessa io* (*Rhi*).

**Figure 8 biology-10-01012-f008:**
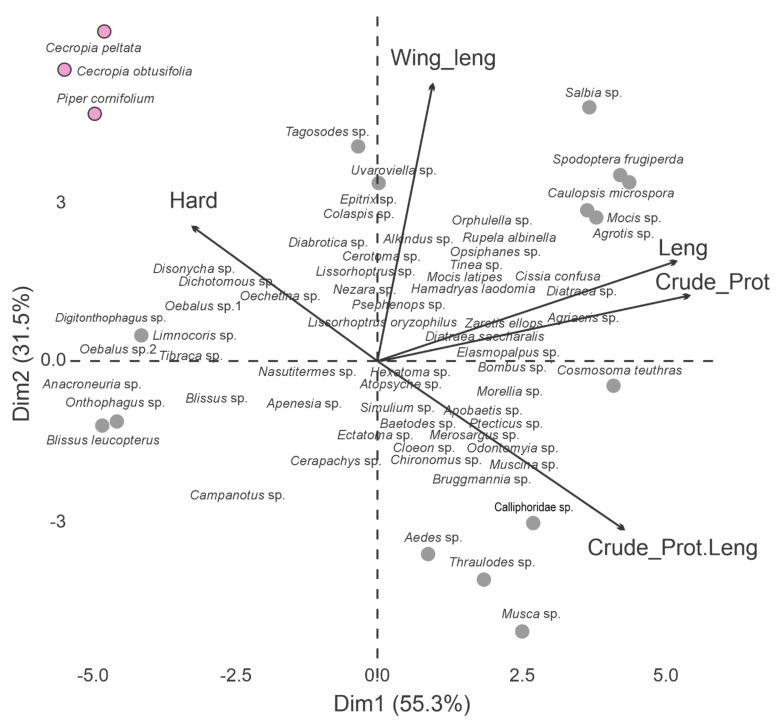
Biplot from a principal component analysis corresponding to the data matrix of the remains of insects and fruits (seeds) consumed by bats and registered in their feces (see also Table A2). Hard: hardness, Leng: length, Crude_Prot: crude protein concentration, Crude_Prot. Leng: ratio of protein concentration length, and Width: width of the wing or fruit. Gray circles: insects, pink circles: seeds of fruits registered in the feces.

**Table 1 biology-10-01012-t001:** Biometric measurements (mm) and bite force (BF) of the skull and jaw recorded from 10 Neotropical species of insectivorous bats. Species, sex, and sample size (n = individuals for each sex) are indicated. The values (pooled data from field collection and voucher specimens) are presented as mean ± SD. The abbreviations for 16 cranial measurements correspond to those in Figure 2. Only the significant intersexual comparisons (MANOVA, Table A3) are shown. *Peropteryx macrotis* (*Pem*), *Saccopteryx bilineata* (*Sab*), *Saccopteryx leptura* (*Sal*), *Noctilio albiventris* (*Noa*), *Molossops temminckii* (*Mot*), *Molossus coibensis* (*Moc*), *Molossus molossus* (*Mom*), *Myotis nigricans* (*Myn*), *Myotis riparius* (*Myr*), *Rhogeessa io* (*Rhi*).

Species	Sex	*Pem*		*Sab*		*Sal*		*Noa*		*Mot*		*Moc*		*Mom*		*Myn*		*Myr*		*Rhi*		Significant ANOVA
n		30		30		23		24		22		24		30		30		30		21		
GLS	F	15.55	±0.28	15.13	±0.07	14.43	±0.30	19.67	±2.39	14.58	±0.21	15.46	±0.23	15.8	±0.41	13.48	±0.78	13.41	±0.81	12.53	±1.27	F = 4.79
	M	15.05	±0.21	15.90	±0.58	13.16	±0.63	19.56	±2.20	13.09	±0.66	15.3	±0.32	15.62	±0.41	13.46	±0.50	13.36	±0.54	11.33	±1.44	*p* < 0.01
CIL	F	13.58	±0.52	13.34	±0.38	12.82	±0.09	16.33	±2.05	12.05	±0.33	12.32	±0.18	13.44	±0.44	11.38	±0.71	11.89	±0.43	9.39	±1.83	F = 12.45
	M	13.37	±0.59	11.92	±0.17	11.63	±0.32	16.39	±2.16	0.73	±0.79	12.1	±0.07	13.9	±0.87	11.31	±0.49	11.63	±0.32	9.45	±1.46	*p* < 0.02
CCL	F	12.35	±0.28	13.06	±0.71	11.70	±0.11	15.28	±2.07	11.12	±0.47	12.02	±0.08	12.83	±0.58	10.55	±0.82	10.44	±0.88	9.5	±1.45	F = 11.60
	M	12.02	±0.39	12.08	±0.42	10.18	±0.62	15.25	±2.16	9.37	±1.07	11.8	±0.27	12.39	±0.59	10.58	±0.40	10.52	±0.44	8.95	±1.30	*p* < 0.02
BB	F	7.09	±0.02	6.07	±0.68	7.30	±0.12	10.55	±2.22	6.89	±0.15	7.89	±0.50	8.11	±0.65	6.23	±0.57	6.22	±0.58	4.79	±1.50	F = 7.00
	M	6.9	±0.01	6.01	±0.54	6.94	±0.03	10.58	±2.26	6.16	±0.45	7.91	±0.62	8.1	±0.74	5.72	±0.71	5.21	±1.03	5.35	±0.94	*p* < 0.04
ZB	F	9.26	±0.05	10.02	±0.31	8.88	±0.23	14.36	±2.33	8.92	±0.21	9.77	±0.19	10.28	±0.43	7.5	±0.87	8.43	±0.43	6.18	±1.48	F = 9.27
	M	8.86	±0.14	9.31	±0.05	8.31	±0.38	15.02	±2.54	7.81	±0.60	9.86	±0.29	10.12	±0.40	7.06	±0.93	8.38	±0.35	7.14	±0.89	*p* < 0.03
MB	F	7.35	±0.06	6.93	±0.34	7.09	±0.23	10.7	±2.10	7.2	±0.2	8.66	±0.78	8.76	±0.84	6.35	±0.71	6.16	±0.84	5.33	±1.37	
	M	7.178	±0.08	6.33	±0.57	6.55	±0.44	11.08	±2.16	6.56	±0.43	8.84	±0.87	8.76	±0.83	6.8	±0.30	5.42	±1.09	5.65	±0.96	
PL	F	6.39	±0.11	6.37	±0.13	6.40	±0.10	9.59	±2.61	6.93	±0.34	5.61	±0.78	6.14	±0.83	6.48	±0.04	6.04	±0.41	5.28	±1.06	
	M	6.13	±0.21	6.39	±0.01	6.15	±0.19	9.59	±2.57	6.18	±0.17	5.52	±0.69	5.96	±0.34	5.19	±0.96	7.15	±0.61	5.60	±0.63	
PB	F	3.00	±0.49	3.21	±0.26	2.45	±1.13	5.57	±2.48	3.62	±0.22	3.57	±0.16	3.41	±0.02	3.46	±0.03	3.51	±0.10	2.48	±1.10	
	M	2.88	±0.56	2.93	±0.51	2.21	±1.29	5.65	±2.41	3.22	±0.21	3.9	±0.53	3.44	±0.03	3.75	±0.36	3.34	±0.08	2.77	±0.69	
MTRL	F	5.95	±0.67	5.65	±0.30	5.72	±0.38	7.18	±2.22	5.14	±0.34	5.09	±0.41	5.32	±0.12	4.98	±0.54	4.94	±0.59	4.17	±1.57	
	M	5.82	±0.77	5.81	±0.77	5.12	±0.10	7.11	±2.01	4.72	±0.29	5.22	±0.19	4.7	±0.31	4.12	±0.86	3.8	−1.16	3.75	±1.21	
M^3^-M^3^	F	6.77	±0.29	6.29	±0.08	6.17	±0.18	9.35	±2.33	6.17	±0.18	6.83	±0.34	7.01	±0.48	5.24	±0.92	5.2	±0.95	4.96	±1.14	
	M	6.66	±0.40	6.46	±0.27	5.85	±0.14	9.34	±2.20	5.46	±0.41	7.06	±0.67	6.21	±0.10	4.89	±0.79	4.48	−1.07	4.23	±1.23	
C-C	F	3.88	±0.09	3.22	±0.48	2.30	±1.30	6.61	±2.49	3.42	±0.31	4.04	±0.23	4.3	±0.46	3.13	±0.57	3.57	±0.18	3.30	±0.41	
	M	3.68	±0.05	3.2	±0.38	2.62	±0.90	6.59	±2.67	3.06	±0.50	3.88	±0.23	3.6	±0.02	3.25	±0.33	3.48	±0.12	2.84	±0.70	
Biometric features (mm) of the jaw																						
DENL	F	10.47	±0.06	11.24	±0.58	10.07	±0.21	13.67	±2.21	10.19	±0.13	10.01	±0.25	11.33	±0.64	9.69	±0.47	8.89	±1.00	8.28	±1.42	F = 10.06
	M	10.13	±0.22	11.07	±0.76	9.27	±0.27	13.2	±1.98	9.01	±0.43	10.38	±0.37	10.31	±0.33	9.42	±0.19	7.61	±1.22	7.02	±1.55	*p* < 0.03
DD	F	1.37	±0.67	1.24	±0.81	1.17	±0.89	3.12	±1.22	2.46	±0.51	3.48	±1.61	2.96	±1.05	1.15	±0.91	1.79	±0.22	1.18	±0.88	
	M	1.23	±0.68	1.07	±0.86	0.97	±0.95	3.22	±1.36	2.39	±0.51	3.45	±1.59	2.81	±0.94	1.09	±0.83	1.62	±0.28	1.12	±0.80	
MFL	F	4.11	±0.02	2.38	±0.78	2.26	±0.84	7.1	±1.40	6.28	±1.02	5.39	±0.61	6.87	±1.29	2.45	±0.75	2.62	±0.67	1.31	±1.28	
	M	4.07	±0.04	2.22	±0.81	2.16	±0.84	7.06	±1.42	6.15	±1.00	5.27	±0.60	6.75	±1.28	2.33	±0.76	2.53	±0.67	1.27	±1.25	
MANDL	F	6.2	±0.06	6.24	±0.09	4.16	±1.84	8.40	±2.10	6.04	±0.09	6.42	±0.26	6.58	±0.41	6.37	±0.22	5.83	±0.29	5.15	±0.91	
	M	6.16	±0.28	6.15	±0.27	6.01	±0.13	7.88	±1.96	5.28	±0.57	6.28	±0.39	5.69	±0.18	6.47	±0.58	4.56	±1.28	4.25	±1.58	
WMC	F	3.15	±0.65	2.76	±0.95	1.89	±1.65	6.01	±1.62	5.01	±0.82	5.06	±0.87	4.93	±0.77	3.57	±0.31	3.85	±0.09	3.44	±0.42	
	M	2.89	±0.87	2.88	±0.88	2.53	±1.20	5.77	±1.80	4.56	±0.68	5.15	±1.23	4.3	±0.43	3.57	±0.24	3.01	±0.76	3.63	±0.18	
BF	F	0.33	±0.31	0.25	±1.00	0.31	±0.54	0.60	±1.79	0.32	±0.42	0.40	±0.17	0.58	±1.65	0.23	±1.12	0.32	±0.42	0.40	±0.21	F = 7.39
	M	0.30	±0.64	0.24	±1.00	0.26	±0.94	0.64	±1.86	0.36	±0.19	0.32	±0.52	0.58	±1.44	0.46	±0.56	0.31	±0.57	0.40	±0.10	*p* < 0.01

**Table 2 biology-10-01012-t002:** Comparison of cranial measurements and bite force (BF) in males and females of insectivorous bats from Colombian tropical dry forests using paired Student t-tests.

Variables	Estimate	Std. Error	t Value	Pr(>|t|)
GLS	14.589	0.205	11.138	<2 × 10^−16^
CIL	−2.346	0.235	−9.992	<2 × 10^−16^
CCL	−3.195	0.235	−13.610	<2 × 10^−16^
BB	−7.794	0.235	−33.199	<2 × 10^−16^
ZB	−5.521	0.235	−23.519	<2 × 10^−16^
MB	−7.410	0.235	−31.564	<2 × 10^−16^
PL	−8.340	0.235	−35.527	<2 × 10^−16^
PB	−11.376	0.235	−48.460	<2 × 10^−16^
MTRL	−9.579	0.235	−40.805	<2 × 10^−16^
M^3^-M^3^	−8.563	0.235	−36.477	<2 × 10^−16^
C-C	−11.096	0.235	−47.268	<2 × 10^−16^
DENL	−4.732	0.235	−20.158	<2 × 10^−16^
DD	−14.414	0.235	−61.403	<2 × 10^−16^
MFL	0.516	0.170	76.030	0.003
MANDL	−8.789	0.235	−37.438	<2 × 10^−16^
WMC	−10.897	0.235	−46.418	<2 × 10^−16^

**Table 3 biology-10-01012-t003:** Correlation analyses using Pearson’s coefficient between bite force (BF) and morphological traits among pairs of species of insectivorous bats from the tropical dry forest in Colombia.

Species	*p* Value	Species	*p* Value	Traits	*p* Value
*M. temminckii-M. coibensis*	0.0156	*M. molossus-S. leptura*	<0.0001	BF-C-C	<0.001
*M. temminckii*-*M. molossus*	0.0006	*M. nigricans-M. riparius*	1.0000	BF-CCL	<0.001
*M. temminckii-M. nigricans*	0.0206	*M. nigricans-R. io*	0.0015	BF-CIL	<0.001
*M. temminckii-M. riparius*	0.0093	*M. nigricans-N. albiventris*	<0.0001	BF-DD	<0.001
*M. temminckii-R. io*	<0.0001	*M. nigricans-P. macrotis*	<0.0001	BF-DENL	<0.001
*M. temminckii-N. albiventris*	<0.0001	*M. nigricans-S. bilineata*	0.0076	BF-GLS	<0.001
*M. temminckii-P. macrotis*	0.7338	*M. nigricans-S. leptura*	0.9918	BF-M^3^-M^3^	<0.001
*M. temminckii-S. bilineata*	1.0000	*M. riparius-R. io*	0.0038	BF-MANDL	<0.001
*M. temminckii-S. leptura*	0.2936	*M. riparius-N. albiventris*	<0.0001	BF-MB	<0.001
*M. coibensis-M. molossus*	0.9981	*M. riparius-P. macrotis*	<0.0001	BF-MFL	<0.001
*M. coibensis-M. nigricans*	<0.0001	*M. riparius-S. bilineata*	0.0032	BF-MTRL	<0.001
*M. coibensis-M. riparius*	<0.0001	*M. riparius-S. leptura*	0.9657	BF-PB	<0.001
*M. coibensis-R. io*	<0.0001	*R. io-N. albiventris*	<0.0001	BF-PL	<0.001
*M. coibensis-N. albiventris*	<0.0001	*R. io-P. macrotis*	<0.0001	BF-WMC	<0.001
*M. coibensis-P. macrotis*	0.7682	*R. io-S. bilineata*	<0.0001	BF-ZB	<0.001
*M. coibensis-S. bilineata*	0.0397	*R. io-S. leptura*	<0.0001	BF-FA	0.425
*M. coibensis-S. leptura*	<0.0001	*N. albiventris-P. macrotis*	<0.0001		
*M. molossus-M. nigricans*	<0.0001	*N. albiventris-S. bilineata*	<0.0001		
*M. molossus-M. riparius*	<0.0001	*N. albiventris-S. leptura*	<0.0001		
*M. molossus-R. io*	<0.0001	*P. macrotis-S. bilineata*	0.8864		
*M. molossus-N. albiventris*	<0.0001	*P. macrotis-S. leptura*	0.0010		
*M. molossus-P. macrotis*	0.2335	*S. bilineata-S. leptura*	0.1583		
*M. molossus-S. bilineata*	0.0021				

## Data Availability

Data are contained within the article or Appendix A. Other data sets used and/or analyzed during the current study are available from the corresponding author or by reasonable request.

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
