# Peer review of "Skull Morphology, Bite Force, and Diet in Insectivorous Bats from Tropical Dry Forests in Colombia"

_biology, 2021, doi:10.3390/biology10101012_

Round 1

Reviewer 1 Report

This is a timely study that aims to add to the knowledge about the effect of bat skull size and morphology on bat choice of prey. It focuses on species of insectivorous bats that occur in the tropical dry forest in Colombia. The ms. is well written and makes the reading pleasant. The methodology is generally adequate, and the results seem to support the results. I believe, however, that there is some room for improvement.

I am adding the ms. pdf file to this review with some comments and suggestions, but my main concerns are:

  • The graphical abstract does not reflect the results of the paper. For example, bat ecosystem services and the importance of bats to the forest (or vice versa) are topics never approach in the ms.
  • There is a clear lack of reference to the Optimal Foraging Theory and other facts that may affect prey selection and use. As a result, the issue is poorly discussed.
  • A major point is the lack of clarity in most sections of the methodology. The authors seem (and I emphasise the 'seem' as I am not sure which variables were included in the models) to use multivariable analysis (like regression and MANOVA) to analyse several parameters that include 20 skull measurements. The absence of multicollinearity is an important assumption of these statistical models that should be taken into consideration. In fact, I see no need of using 20 skull measurements, not only in multivariate analysis, as most of them will only add noise to the study and increase its complexity. Instead, I recommend a process of variable reduction before the statistical analysis, excluding correlated/redundant variables.
  • Without a clear description of the statistical procedure, I cannot be confident of the robustness of your results.
  • Although the Journal Biology's policy of accepting long papers with very detailed statistical analysis, I think that there is some redundancy in the analysis performed and that the author's hypothesis could be tested, and the results would be much clearer, with less analysis and statistical procedures. I would like to highlight that I could not see both Figures 4 and 6, despite opening the file on different computers and Adobe Acrobat versions.

Below I am adding a list of suggestions and comments made to the ms. These were extracted from the pdf file that I am attaching to this review, and I think it will be simpler for the authors to consult the file.

Page 1

Author: Rev Subject: Sticky Note Date: 21/05/2021 10:48:53 Length of the skull, of the jaw? Please clarify.

Page 3

Author: Rev Subject: Highlight Date: 21/05/2021 10:47:35

Author: Rev Subject: Sticky Note Date: 21/05/2021 10:53:30 This is the only place where I found the mention to ecosystem services in the ms. It is thus strange it would appear in the abstract (even if graphical)

Author: Rev Subject: Sticky Note Date: 21/05/2021 10:55:39 Again, I found no reference in the ms. on how the relationship between BF and skull morphology and the importance of the tropical dry forest to bats

Author: Rev Subject: Sticky Note Date: 21/05/2021 11:22:43 Not clear what you mean by this, but I would say that 'simple anatomy' oversimplifies a complex issue. Perhaps you could rephrase it.

Author: Rev Subject: Sticky Note Date: 21/05/2021 11:47:16 Factors other than BF may be at play, such as prey availability, capture success and digestibility. Optimal foraging theory would say that if bats have abundant and easy to capture soft prey, they will not waste time and energy searching for hard prey just because they have the bite force to eat them.

Page 4

Author: Rev Subject: Sticky Note Date: 21/05/2021 12:48:00 Please check if listing the municipalities is important for your results.

Author: Rev Subject: Sticky Note Date: 21/05/2021 13:38:35 Please clarify the relationship between the fruits consumed by N. albiventris (identified by the seeds in faeces) and the fruits of consumption you used to measure fruit hardness. Are they the same, only part or no relation between them?

Author: Rev Subject: Sticky Note Date: 21/05/2021 13:41:07 Not clear. You related fruit hardness to each individual plant species. Where there plant species that would grow more than one fruit type?

Author: Rev Subject: Sticky Note Date: 21/05/2021 14:02:24 Did you capture and collect faeces of all these species during your fieldwork? Perhaps you could say which species were caught with mistnets in the previous section.

Author: Rev Subject: Sticky Note Date: 21/05/2021 13:56:15 Please consider if the list is necessary.

Author: Rev Subject: Sticky Note Date: 21/05/2021 14:07:29 Did you check for correlation/collinearity between measurements before making the regression? I would say that at least some of the 20 measurements are correlated; non-collinearity between descriptors is an important assumption of regression that you should check and mention the results.

Author: Rev Subject: Sticky Note Date: 21/05/2021 14:18:24 Perhaps you should have used mixed models instead, with FA and GLS as random factors. I trust this would be the appropriate way to do it.

  1. Perhaps I understood it wrong. The regression did not include the skull variables? If that is the case, you should remove the first three lines or place the sentence in a different paragraph.

Author: Rev Subject: Sticky Note Date: 21/05/2021 14:13:34 Again, the absence of multicollinearity is one of the assumptions of MANOVA. Please explain how did you cope with that.

Author: Rev Subject: Sticky Note Date: 21/05/2021 14:14:53 Please clarify. The skull measurements were not considered in the models?

Page 6

Author: Rev Subject: Inserted Text Date: 21/05/2021 14:21:57 between bat species

Author: Rev Subject: Sticky Note Date: 21/05/2021 14:24:08 Please relate to what you wrote in line 183. How does this analysis relate to the MANOVA performed before?

Author: Rev Subject: Sticky Note Date: 21/05/2021 14:25:59 Please add information on the procedure taken when you found correlated descriptors

Author: Rev Subject: Sticky Note Date: 21/05/2021 14:28:11 Please add information on what samples were paired

Author: Rev Subject: Sticky Note Date: 21/05/2021 14:27:25 Graphical representation of what? Please clarify.

Author: Rev Subject: Sticky Note Date: 21/05/2021 14:31:31 It is usually good practice to list the packages used.

Page: 7

Author: Rev Subject: Sticky Note Date: 21/05/2021 15:23:59 please clarify to what morphology you refer to or what you mean simply by skull. Could it be "skull morphology"? Author: Rev Subject: Cross-Out Date: 21/05/2021 15:20:33

Page: 8

Author: Rev Subject: Sticky Note Date: 21/05/2021 15:27:46 You say that species are different, which is not very informative. Please clarify on what they differ: BF, skull shape?

Author: Rev Subject: Sticky Note Date: 21/05/2021 15:32:32 Please consider clarifying in what males and females did not differ. Please relate with previous statements: Ln 245 - "significant differences between sexes" [in skull morphology] Ln 270 - "Females have stronger BF than males"

Author: Rev Subject: Sticky Note Date: 21/05/2021 14:43:05 Very poor quality image. I cannot read the coordinates at the border of the map or what it is inside the box at the bottom left corner.

I think Scale is written as Escala (or similar)

Author: Rev Subject: Sticky Note Date: 21/05/2021 14:42:54 Could not relate this colour to any on the legend

Author: Rev Subject: Sticky Note Date: 21/05/2021 14:47:58 Please consider if it is relevant to identify each department this way. Perhaps you could distinguish the sites you sampled and the sites with voucher specimens and add some department names to the map. The habitat is only visible where you have no hexagons, so it is also not so informative. Perhaps you could use empty symbols in each site (only with a border).

Page: 11

Author: Rev Subject: Sticky Note Date: 21/05/2021 15:12:23 This figure is not visible. Tried in different pcs.

Author: Rev Subject: Sticky Note Date: 21/05/2021 15:12:23 I marked the pairs of variables that you should consider not to include simultaneously in the same model. I count 7 variables, including BF Author: Rev Subject: Highlight Date: 21/05/2021 15:13:57

Page: 12

Author: Rev Subject: Sticky Note Date: 21/05/2021 15:03:25 Again, the Figure is not visible.

Page: 13

Author: Rev Subject: Sticky Note Date: 21/05/2021 15:47:24 Please add the fruit/insect colour code.

Page: 18

Author: Rev skull?       Subject: Inserted Text             Date: 21/05/2021 15:52:42

Author: Rev skull         Subject: Inserted Text             Date: 21/05/2021 16:00:18

Author: Rev skull         Subject: Inserted Text             Date: 21/05/2021 15:56:16

Page: 19

Author: Rev Subject: Sticky Note Date: 21/05/2021 16:06:17 A bit out of context. Please consider rephrasing.

Author Response

Reviewer 1

Comments and Suggestions for Authors

This is a timely study that aims to add to the knowledge about the effect of bat skull size and morphology on bat choice of prey. It focuses on species of insectivorous bats that occur in the tropical dry forest in Colombia. The Ms. Is well written and makes the reading pleasant. The methodology is generally adequate, and the results seem to support the results. I believe, however, that there is some room for improvement.
R/: Thank you for your comments and your inputs to improve our manuscript.

1. Page 1. Lines 43. Length of the skull, of the jaw? Please clarify.
R/: We included this suggestion, please see now the lines 42-43.

2. Page 3. Lines 51. Graphical abstract. This is the only place where I found the mention to ecosystem services in the Ms. It is thus strange it would appear in the abstract (even if graphical). Again, I found no reference in the ms. on how the relationship between BF and skull morphology and the importance of the tropical dry forest to bats.
R/ We agree with this comment, and we have removed these two comments from the graphical abstract and we have now added a new graphical summary for attending the suggested changes. See Line 52.

3. Page 3. Lines 63-64. Introduction. Not clear, what you mean by this, but I would say that 'simple anatomy' oversimplifies a complex issue. Perhaps you could rephrase it.
R/: Yes, we agree with this suggestion, and we have rephrased it paragraph. Please check the lines 66-69.

4. Page 3. Lines 72-74. Introduction. Factors other than BF may be at play, such as prey availability, capture success and digestibility. Optimal foraging theory would say that if bats have abundant and easy to capture soft prey, they will not waste time and energy searching for hard prey just because they have the bite force to eat them.
R/: Thank you, we agree with this comment, and have added a new paragraph that includes this suggestion, supported by at least two additional references. Please check the lines 79-86.

5. Page 4. Lines 90-92. Materials and Methods. Please check if listing the municipalities is important for your results.
R/ Thank you for your suggestion. We deleted the. Please see the lines 93-97;104-105. It is also visible in the new version of the Figure 1 (map).

6. Page 5. Lines 137. Materials and Methods. Please clarify the relationship between the fruits consumed by N. albiventris (identified by the seeds in faeces) and the fruits of consumption you used to measure fruit hardness. Are they the same, only part or no relation between them?
Page 5. Lines 138- 139. Materials and Methods. Not clear. You related fruit hardness to each individual plant species. Where there plant species that would grow more than one fruit type?
R/: Thank you for this key detail. It was an unintentional mistake from the original version of our MS. Now, we had added a phrase for details related to seeds belonging to fruits of three plant species that were registered in the faeces o f N. albiventris and they are present in the studied areas of the Colombian TDF. Please check the lines 147-153.

7. Page 5. Lines 147-149. Materials and Methods. Skull Morphology. Did you capture and collect faeces of all these species during your fieldwork? Perhaps you could say which species were caught with mistnets in the previous section.
R/ Thank you for your suggestion. We added the two requested affirmations. Please see the lines 108 and 135-136.

8. Lines 153-154. Materials and Methods. Skull Morphology. Please consider if the list is necessary.
R/ We agree with this suggestion, and we have deleted these issues. Please see the lines 162-163.

9. Page 5. Lines 165. Materials and Methods. Statistical Analyses. Did you check for correlation/collinearity between measurements before making the regression? I would say that at least some of the 20 measurements are correlated; non-collinearity between descriptors is an important assumption of regression that you should check and mention the results.
R/: Thank you for these observations. In the first sentence of the Statistical Analyses subsection, we tested for collinearity and discarded the variables MLTRL, M1-M1, MXBR, and COH.

10. Page 5. Lines 172-173. Materials and Methods. Statistical Analyses. Perhaps you should have used mixed models instead, with FA and GLS as random factors. I trust this would be the appropriate way to do it.
R/: Yes, we agree with this suggestion, and have added a new paragraph with this issue. Please see the lines 179-192.
Page 5. Perhaps I understood it wrong. The regression did not include the skull variables? If that is the case, you should remove the first three lines or place the sentence in a different paragraph.
R/: Thank you for your suggestion. We delete the confounding words and adjust the paragraph. Please check the line 200-202.

11. Lines 180-182. Materials and Methods. Variation in Skull Morphology. Again, the absence of multicollinearity is one of the assumptions of MANOVA. Please explain how did you cope with that.
R/: Thank you very much for raising this important issue. Please, check above our response to the 9-comment. By reanalyzing our data, the variable M3-M3 was included and M2-M2discarded. Please see the lines 186-190.

12. Lines 183-184. Materials and Methods. Variation in Skull Morphology. Please clarify. The skull measurements were not considered in the models?
R/: Thank you for your suggestion. We discarded the variable GLS, because it was previously included in our statistical analysis and include the variable body mass as a control for potential confounding effects related to body size or biometric relationships. Please see the lines 200-202.

13. Page 6. Lines 186-187. Between bat species.
R/: Thank you for your suggestion. We adjusted the terms "Between bat species". Please see the lines 211-212.

14. Page 6. Lines 203. Materials and Methods. Bite Force Between Bat Species. Please relate to what you wrote in line 183. How does this analysis relate to the MANOVA performed before?
R/: Thank you very much for raising this important issue. Please check our response above in the 11-comment and see the lines 186-190, and 206-207.

15. Page 6. Lines 211. Materials and Methods. Bite Force, Skull Morphology, and Diet of Bats. Please add information on the procedure taken when you found correlated descriptors.
R/: Thank you very much for raising this important issue. Please check the above response in the 9-comment and see the lines 238-242.

16. Page 6. Lines 212-213. Materials and Methods. Bite Force, Skull Morphology, and Diet of Bats. Please add information on what samples were paired
R/: Yes, we agree with this suggestion, and we have added a new paragraph. Please see the lines 242-245.

17. Page 6. Lines 213. Materials and Methods. Bite Force, Skull Morphology, and Diet of Bats. Graphical representation of what? Please clarify.
R/: We adjusted the phrase. Please see the lines 245-246.

18. Page 6. Lines 222. Materials and Methods. Bite Force, Skull Morphology, and Diet of Bats. It is usually good practice to list the packages used.
R/: Thank you for this key detail. We added the suggested phrase. Please see the lines 254-257.

19. Page: 7. Lines 266. Materials and Methods. Bite Force Between Bat Species. Please clarify to what morphology you refer to or what you mean simply by skull. Could it be "skull morphology"?
R/: Thank you for this key detail. This was clarified, please see the lines 300-301.

20. Page: 8. Lines 282. Materials and Methods. Bite Force Between Bat Species. You say that species are different, which is not very informative. Please clarify on what they differ: BF, skull shape?
R/: Thank you for your suggestion. This was clarified, please see the lines 320-323.

21. Lines 283. Materials and Methods. Bite Force Between Bat Species. Please consider clarifying in what males and females did not differ. Please relate with previous statements: Ln 245 - "significant differences between sexes" [in skull morphology] Ln 270 - "Females have stronger BF than males"
R/: Thank you for your suggestion and specific detail. We had clarified these issues. Please see the lines 280-282, 306-308, and 323-324.

22. Lines 303. Materials and Methods. Figures, Tables and Schemes. Very poor quality image. I cannot read the coordinates at the border of the map or what it is inside the box at the bottom left corner. I think Scale is written as Escala (or similar). Could not relate this colour to any on the legend. Please consider if it is relevant to identify each department this way. Perhaps you could distinguish the sites you sampled and the sites with voucher specimens and add some department names to the map. The habitat is only visible where you have no hexagons, so it is also not so informative. Perhaps you could use empty symbols in each site (only with a border).
R/: Thank you for pointing this out. We included a new and edited map (Figure 1). We corrected the term to Scale and put more clear descriptors for samples and sampling sites in the map and using empty square and circle symbols. Please see the line 345.

23. Page: 11. Lines 339. Figure 4. This figure is not visible. Tried in different pcs.
R/: Thank you for the suggestion. We had included (newly) the Figures 4 and 6. These also will be upload independently. Please see the line 374.

24. Lines 347. I marked the pairs of variables that you should consider not to include simultaneously in the same model. I count 7 variables, including BF
R/: Thank you for this key suggestion. Based on the reanalyzes of the data, now we are highlight only 6 relevant variables of 16, that were associated to the BF. Please see the line 383 and the edited Figure 5.

25. Page: 12. Lines 347. Figure 6. Again, the Figure is not visible.
R/: Thanks for the suggestion. Please see the line 391, and the uploaded Figure 6.

26. Page: 13. Lines 380. Figure 8. Please add the fruit/insect colour code.
R/: Thank you for the suggestion. It was done. Please see the line 415, and edited Figure 8.

27. Page: 19. Lines 483-485. Discussion. A bit out of context. Please consider rephrasing.
R/: Thank you for the suggestion. Please check now the lines 518-519.

Reviewer 2 Report

After reading the manuscript “Skull morphology, bite force and diet in insectivorous bats from a tropical dry forest in Colombia” by Ramírez-Fráncel et al., I am convinced that this study contains information of interest. However, it needs to be modified with a few changes to be improved.

Minor comments:

Line 41: Please, rephrase or clarify this sentence and add the reference.

Figure 1: Please, improve the quality of the image.

Line 135: Please, write the genera of “N. albiventris”, it is the first time you mention it so it shouldn´t be abbreviated. Control these details through all the manuscript.

Lines 146-149: I think it would be better if the authors included this information in a table, incorporating the number of individuals examined, females and male per bat species and maybe where they were collected.

Lines 165-167: As I mentioned before, I believe the authors should include the number of individuals, females and male per species of bat.

Table 1: I think this table could be incorporated in the appendix section.

Table 2: This table is not cited in the text.

Figure 7: Please abbreviate the names of the bat species.

Line 489-491: For a better and resent reference please cite the review entitled:

“Salinas-Ramos, V. B., Ancillotto, L., Bosso, L., Sánchez‐Cordero, V., & Russo, D. (2020). Interspecific competition in bats: state of knowledge and research challenges. Mammal Review, 50(1), 68-81.”

Author Response

Reviewer 2

Comments to the Author

After reading the manuscript “Skull morphology, bite force and diet in insectivorous bats from a tropical dry forest in Colombia” by Ramírez-Fráncel et al., I am convinced that this study contains information of interest. However, it needs to be modified with a few changes to be improved.

R/: Thank you for your comment and for highlighting the value of our manuscript.

  1. Line 41. Please, rephrase or clarify this sentence and add the reference.

R/: Thank you for this comment. We included more clear ideas and the reference by Senawi et al. (2015). Please see the lines 41 to 42.

  1. Figure 1: Please, improve the quality of the image.

R/ Thank you for your suggestion. We corrected the term to as Scale and edited the map with clearer internal descriptors. Please see the line 345 and the edited Figure 1.

  1. Line 135. Please, write the genera of “N. albiventris”, it is the first time you mention it so it shouldn´t be abbreviated. Control these details through all the manuscript.

R/. Thank you. We agree with this rule, and we have now added the genus for the species Noctilio albiventris. Please see the line 144.

  1. Lines 146-149. I think it would be better if the authors included this information in a table, incorporating the number of individuals examined, females and male per bat species and maybe where they were collected.

R/ Thank you for your suggestion. We highlight this issue for citing the appendix (Appendix I). Please see the line 159.

  1. Lines 165-167. As I mentioned before, I believe the authors should include the number of individuals, females and male per species of bat.

R/ Thank you for your suggestion. We cite the appendix that include this information. Please see the line 192.

  1. Table 1: I think this table could be incorporated in the appendix section.

R/: Thank you for this key detail. However, respectfully we think this table contain valuable information that extracting it from the manuscript would take away much value from the final version of the paper.

  1. Table 2: This table is not cited in the text.

R/: Thank you for this key detail. Now, we cite this table in the text in the line 245. 

  1. Figure 7: Please abbreviate the names of the bat species.

R/ Thank you for your suggestion. We adjusted the Figure 7 attending to your comment.

  1. Line 489-491. For a better and resent reference please cite the review entitled:

“Salinas-Ramos, V. B., Ancillotto, L., Bosso, L., Sánchez‐Cordero, V., & Russo, D. (2020). Interspecific competition in bats: state of knowledge and research challenges. Mammal Review, 50(1), 68-81.”

R/: Thank you. We include the reference by Salinas-Ramos et al. (2020). Please see the line 526.

Round 2

Reviewer 1 Report

Nicelly done, good job.